# Molybdenum as a Potential Biocompatible and Resorbable Material for Osteosynthesis in Craniomaxillofacial Surgery—An In Vitro Study

**DOI:** 10.3390/ijms232415710

**Published:** 2022-12-11

**Authors:** André Toschka, Georg Pöhle, Peter Quadbeck, Christoph V. Suschek, Alexander Strauß, Christian Redlich, Majeed Rana

**Affiliations:** 1Department of Oral-, Maxillofacial and Facial Plastic Surgery, University Hospital Düsseldorf, 40225 Düsseldorf, Germany; 2Fraunhofer Institute for Manufacturing Technology and Advanced Materials IFAM, Branch Lab Dresden, 01277 Dresden, Germany; 3Department of Electrical Engineering, Medical Engineering and Computer Science (EMI), Offenburg University, 77652 Offenburg, Germany; 4Department for Orthopedics and Trauma Surgery, Medical Faculty, Heinrich-Heine-University Düsseldorf, 40225 Düsseldorf, Germany

**Keywords:** biomaterials, craniomaxillofacial surgery, osteosynthesis, biodegradable implants, molybdenum

## Abstract

Titanium and stainless steel are commonly known as osteosynthesis materials with high strength and good biocompatibility. However, they have the big disadvantage that a second operation for hardware removal is necessary. Although resorbable systems made of polymers or magnesium are increasingly used, they show some severe adverse foreign body reactions or unsatisfying degradation behavior. Therefore, we started to investigate molybdenum as a potential new biodegradable material for osteosynthesis in craniomaxillofacial surgery. To characterize molybdenum as a biocompatible material, we performed in vitro assays in accordance with ISO Norm 10993-5. In four different experimental setups, we showed that pure molybdenum and molybdenum rhenium alloys do not lead to cytotoxicity in human and mouse fibroblasts. We also examined the degradation behavior of molybdenum by carrying out long-term immersion tests (up to 6 months) with molybdenum sheet metal. We showed that molybdenum has sufficient mechanical stability over at least 6 months for implants on the one hand and is subject to very uniform degradation on the other. The results of our experiments are very promising for the development of new resorbable osteosynthesis materials for craniomaxillofacial surgery based on molybdenum.

## 1. Introduction

Despite the development of various non-resorbable and resorbable materials for osteosynthesis, there remains high interest and the requirement for new bioresorbable materials with high strength and excellent biocompatibility.

Osteosynthesis materials made from non-resorbable metal alloys demonstrate sufficient biocompatibility and great stability. Titanium implants are considered as the gold standard in osteosynthesis materials. Although titanium is generally highly biocompatible, some severe adverse reactions such as allergies and immune reactions were described [1,2]. Especially in pediatric surgery, these non-resorbable materials have the big disadvantage of the necessity of a second operation for hardware removal, which is usually recommended 6–12 month after first operation. This second operation is associated with high physical and psychological stress for the young patients. Beside the risk of general complications such as infection and a second general anesthetic, hardware removal can also lead to growth arrest.

Especially for young patients such as children in development, there is a high need to implement bioresorbable materials for osteosynthesis. Various resorbable materials are increasingly used with high success [3]. Today, there are the two big groups, namely magnesium-based implants, and polymer-based implants.

Magnesium-based bioresorbable implants have been known for several decades now. Although the material is now used in orthopedics and cardiology, the uneven corrosion of magnesium is a persistent problem. In particular, pitting corrosion frequently leads to undesirable results [4]. In addition, magnesium has a rather low mechanical strength, which must be compensated for in the design of implants by sufficient component thickness. In many cases, such a thickness is no longer appropriate for the corresponding implant.

Most polymer-based implants are made of polylactic acids [5], and they are designed to degrade in vivo via hydrolysis. The first of these systems for facial bone surgery have been available since the late 1980s [6]. The big challenge and limitation lie in the balance between rate and speed of degradation and maintenance of (mechanical) stability. Plates made of the L-isomer of PLA (PLLA) possess high strength and durability because of its hydrophobic, crystalline nature, which increases resistance to degradation [7]. However, PLLA has a propensity to induce foreign body reactions, including sterile abscess formation, which often makes hardware removal necessary [8].

On the other hand, polyglycolic copolymers (PGA), which are more hydrophilic, tend to degrade too quickly and lack the appropriate strength to support bone healing, even in non-load-bearing areas such as the craniofacial skeleton. Therefore, most current plating systems attempt to combine the positive qualities of both materials and are copolymers of PGA and PLA (PLGA) such as RapidSorb^®^ (DePuy Synthes CMF, West Chester, PA, USA), LactoSorb^®^ (Biomet Inc., Jacksonville, FL, USA) or Resorb-X^®^ (KLS Martin, Tuttlingen, Germany) (PDLLA), which are frequently used in craniofacial surgery [9]. Although these resorbable systems for osteosynthesis are increasingly used, foreign body reactions are consistently described [10]. Furthermore, these osteosynthesis systems lack stability in comparison to titanium [11].

These facts suggest that there is still a need for new bioresorbable materials with high biocompatibility and high mechanical stability.

One potentially interesting material is tungsten. Before the late 1990s, coils made of pure tungsten wire were used to treat intracranial aneurysms. However, starting with an article by Weill et al. in 1998, observations on the loss of radiopacity of implanted tungsten coils over 1–3 years were published [12,13,14]. Kampmann et al. showed a total loss of radiographic visibility, indicating total degradation, within 3 years for 60% of implanted tungsten coils [15]. Consequently, aneurysm coils made of tungsten were withdrawn from the market. Subsequent studies showed marked degradation and cracking of tungsten coils in a rabbit model within 4 months [16]. Immediate negative effects of dissolved tungsten in the blood serum, such as systemic toxicity, were not observed in patients or animal studies. The in vitro cytotoxicity was established to be low over the short term [16,17], raising the possibility of using tungsten for biodegradable implants. However, tungsten does not occur physiologically in the human body, raising concerns about long-term biocompatibility. A steady increase in the tungsten level in the blood serum in the studies points towards slow physiological regulation of the tungsten level. Regarding mechanical properties, the low ductility of pure tungsten [18,19] limits options for implant manufacture and use for implants under high loads.

Molybdenum is a trace metal which is involved in and required for multiple enzymatic processes in the human body [20,21]. In recent publications, it is described as a bioresorbable material with high strength and exceptional mechanical properties [5,22].

Since molybdenum and tungsten are Group VI elements with the same valence electron configuration, they are chemically similar. Consequently, molybdenum was also shown to dissolve in various physiological salt solutions in vitro, with a degradation rate of 0.6–18 µm/y depending on the solution used and the form of the investigated samples [23,24]. In these studies, no pitting corrosion occurred. However, regarding use as a degradable biomaterial, molybdenum has two advantages over tungsten. First, molybdenum is a homeostatically regulated essential element in the human body [20]. The total amount of molybdenum in the human body is 10–15 mg. Molybdenum levels are regulated by renal excretion. Increased molybdenum uptake leads to increased renal clearance [25]. Based on a No Observed Adverse Effect Level of 0.9 mg/d per kg body weight, a recommended dietary Upper Intake Level of 0.6–2 mg/d was defined for humans [26]. Molybdenum was shown to be neither mutagenic, cancerogenic or teratogenic [27,28]. In vitro, thrombogenicity and inflammatory response (THP-1 monocytes) were shown to be similar to 316L stainless steel, while toxicity to coronary artery endothelial cells and coronary artery smooth muscle cells was shown to be low [29]. It is therefore expected that the long-term biocompatibility of a degradable molybdenum implant is better than in the case of tungsten. Second, pure molybdenum with a fine-grained microstructure is much more ductile than tungsten. As a measure of ductility, the elongation to failure can reach 40–50%. The control of impurities in the material such as oxygen is essential for good mechanical properties [30]. There is thus lower concern about brittle failure of a molybdenum implant under high loads.

In vitro and in vivo studies have shown that the degradation products consist of molybdenum oxides and calcium phosphates. When molybdenum comes into contact with simulated body fluid or body fluid, it undergoes a cascade of reactions that lead to the dissolvable oxide MoO_3_. This oxide dissolves in the form of molybdate ions MoO_4_^2−^ which can be excreted renally [31,32]. While complete absorption has not yet been demonstrated in vivo, in vitro experiments indicate that molybdenum will dissolve completely in a physiological environment [29,33].

Based on these insights, we considered molybdenum as a promising material for osteosynthesis.

The aim of our experiments was to show the in vitro biocompatibility of pure molybdenum and its alloys with different types of cells in accordance with ISO-Norm 10993-5. Furthermore, we performed long time in vitro experiments to evaluate the mechanical performance and the degradation rate up to 6 months. With our study, we want to take the first step towards a more detailed characterization of this potential new resorbable material for osteosynthesis.

## 2. Results

In a total of four different assays, we repeatedly showed that molybdenum and molybdenum rhenium alloys do not cause significant cytotoxicity in either human or mouse skin fibroblasts nor human osteoblasts. Addition of rhenium increases the ductility [30]. Because of that fact, we also decided to test molybdenum-rhenium alloys, which is an alloy already commercially available. However, rhenium is very expensive and not easily available, which makes its clinical use seem rather unlikely and served more for comparison purposes here.

We seeded primary human skin fibroblasts and primary human osteoblasts at a density of 2 × 10^5^ cells on a cast molybdenum crucible with an internal diameter of 4.4 cm and compared their growth and vitality after 24 h in comparison to corresponding cell cultures maintained on a plastic cell culture plate under otherwise identical conditions.

As can be seen from Figure 1A, both human cell types do not show any statistically significant differences in terms of growth and vitality compared to the cells cultivated in the plastic cell culture plates. In Figure 1B, we show that fluoresceine diacetate (DFA) stained, i.e., living human osteoblasts can grow on the molybdenum plate and show the morphology typical of osteoblast cultures.

We then performed the Cell titer blue assay with L929 mouse fibroblast cells. Two different charges (SRTPO80718 and PDMPB1350) of pure molybdenum powder with high purity (>99.5% purity) and molybdenum rhenium alloys with 48% and 41% of rhenium (Mo48Re and Mo41Re) did not lower the cell viability in comparison to the control. The positive control with H_2_O_2_ showed the functionality of the attempt (Figure 2). A *p*-value of <0.05 as compared to the non-treated control was considered significant.

In Figure 3, the results of the MTT assay are shown. We used the same materials as in the cell titer blue assay and measured the cell viability of L929 mouse fibroblasts via MTT assay. We did not observe any significant differences in rates of cell viability. As a positive control, we again used H_2_O_2_ and saw a statistically significant reduction in cell viability as compared to the non-treated control cultures.

We measured the cell viability with the same compounds in the neutral red assay as well (Figure 4). The same positive and negative controls were used. In the neutral red assay, no significant difference in the amount of living cells was observed compared to the negative control. In contrast, maintaining the cultures in the presence of H_2_O_2_ showed a statistically significant reduction in the neutral red signal as compared to the non-treated controls.

Figure 5 shows SEM (scanning electron microscope) images of metallographic cross-sections of molybdenum sheet metal after 2, 4 and 6 months (from left to right, for both grain orientations) of immersion in Kokubo’s SBF (simulated body fluid). For a detailed description of the grain structure, please see Section 4. The growth of a degradation product layer with uniform thickness on the surface of the metallic sheet is observed, confirming the behavior described in earlier reports. No significant difference in layer thickness is observed in sheets of either parallel or perpendicular grain orientation. Cracks in the degradation product layer and a gap between the layer and the metallic surface are apparent. However, the close fit of the fractured parts and the fact that no new degradation product layer has formed beneath the monolithic layer shows that the cracks do not form in situ, but after removal of the samples from the immersion medium and drying. The cracking of the degradation product layer during drying has been illustrated in the Supplementary Information of Schauer et al. [33].

EDS (energy dispersive X-ray spectroscopy) analysis (Figure 6, left image and table) and EDS mapping (Figure 6, right column) show that the metallic sheet is unchanged and consists of pure molybdenum. The degradation product layer consists mostly of molybdenum, oxygen, calcium, and phosphorus. Minor quantities of magnesium, sodium and potassium are also present. This is consistent with earlier publications which reported degradation product layers consisting of molybdenum oxide and calcium phosphate. Based on this observation, calculating the loss of molybdenum from the molybdenum sheet by comparing the weight before and after immersion is not expedient. It is more useful to calculate total molybdenum dissolution based on ICP-OES (inductively coupled plasma optical emission spectrometry) data of the molybdenum concentration in used immersion medium.

Area-specific molybdenum mass loss, calculated based on the ICP-OES data, is shown in Table 1. Approximately 20% of the original mass of the Mo sheet is lost by dissolution over 6 months. No significant difference in mass loss was observed for parallel and perpendicular grain orientation.

Therefore, the average of all values was calculated and plotted over immersion time (Figure 7). Mass loss over time can be approximated by the fit function Δm = 0.034 t + 0.0001 t^2^. The total dissolution rate after 6 months calculated from the mass loss is ca. 25 µm/year.

Figure 8 shows the change in thickness of the metallic portion of the sheet over time (left). The thickness of the metallic portion declines by around 25 µm over 6 months (25 µm/year over 6 months for the two exposed large surfaces, with the much smaller surfaces of the edges neglected). However, the total thickness of the samples is not reduced by 25 µm since the degradation layer forms on the sample and takes up some of the volume of the original material. The degradation product layer grows in a roughly linear way, as shown in Figure 8 (right), while the reduction of the metallic part can be approximated by an exponential function. Based on these observations, it is predicted that the metallic part of the molybdenum sheet would disappear after approx. 10 months, while the total thickness is expected to decrease more slowly. The degradation products will persist for some time after all of the metal is dissolved or degraded.

Figure 9 (left) shows the tensile strength *R_m_* of the samples as received and after immersion for 2, 4 and 6 months. Note that the values of the tensile strength in MPa are calculated based on the original cross-section, not the cross-section after immersion, to demonstrate how the strength of an implant would change over time. There is no significant difference between the strength of samples with different grain orientation. Over 6 months, the tensile strength of the material is reduced by approx. 50%. Following the exponential decrease in metallic thickness, the loss of tensile strength can be approximated by an exponential function, just like the decrease in metallic thickness. However, due to the high mechanical strength of molybdenum, a tensile strength of more than 400 MPa relative to original cross section remains after 6 months, fulfilling the requirements for most load-bearing orthopedic implant applications. Based on the decrease in the thickness of the metallic portion of the sample, a total loss of mechanical strength is expected after 7.5–9 months. For this estimate, a negligible mechanical strength of the degradation product layer is assumed. The plot of the yield strength *R_p_*_0.2_ over time in Figure 9 (right) follows a similar progression, with a reduction of around 50% in 6 months and an expected total loss of yield strength within the same timeframe as for *R_m_*.

## 3. Discussion

Overall, our results are in agreement with recent published results of Redlich et al. and Sikora-Jasinska et. al., which showed that the degradation behavior and biocompatibility of molybdenum is promising for future cardiovascular stents [5,29].

Naturally, the requirements for cardiac stents and osteosynthesis materials differ in many respects. In particular, the dimensions and forms required are very different—plates and wires in osteosynthesis compared to the stents with fine strut networks in cardiovascular applications. However, as this work shows, molybdenum has great potential for osteosynthesis material in addition to the potential for cardiovascular stents.

Uniform degradation of pure molybdenum sheet metal under simulated physiological conditions (Kokubo’s SBF, pH 7.4, 37 °C) was demonstrated. Over 6 months, a mass loss of molybdenum of approx. 20%, corresponding to a decrease in metal sheet thickness of approx. 25 µm, was observed. This rate of degradation shows that there is a reasonable expectation that a molybdenum-based implant will degrade within a clinically relevant timeframe, e.g., 1–2 years. At the observed mass loss rates, the metallic portion of a 95 µm molybdenum sheet is expected to last for around 10 months. The degradation products are expected to persist for a longer time than the metal. Mechanical testing shows the high potential for use of molybdenum in load-bearing orthopedic implants, with tensile strength and yield strength exceeding 350 MPa after 6 months of degradation. This is still higher than the requirements for most orthopedic applications. For example, for wires used in sternal closure, a yield strength of >300 MPa is a common benchmark.

In addition to the promising degradation measurements, we were able to demonstrate that pure molybdenum and alloys with 48% and 41% rhenium have no significant effects on the viability of mouse and human fibroblasts, as well as on human osteoblasts, which is the basic requirement for further in vivo experiments. Our experiments were conducted as described in ISO standard 10993-5, which is also used for the approval of medical devices according to the Medical Device Regulation (Regulation (EU) 2017/745), especially implants.

In the next steps, the in vivo biocompatibility and the exact degradation rates in complex organisms must be investigated. While Sikora-Jasinska et al. showed the biocompatibility and degradation behavior in a flowing system [5] and Schauer et al. demonstrated degradation in the abdominal aorta of rats [33], the behavior as an osteosynthesis system should be investigated separately. In particular, the degradation and dissolution behavior of molybdenum sheet metal should be investigated under in vivo conditions consistent with implant applications in craniomaxillofacial surgery.

In further in vivo experiments, the main questions will be how the degradation behaves for molybdenum samples of adequate dimensions, how the in vivo environment (e.g., oxygen partial pressure in tissues and fluid exchange rates) changes the degradation behavior and how the degradation products affect complex organisms. The main aspects of this will be the exact degradation rates and the gradual reduction in mechanical stability. Furthermore, it is essential to know in which form the mammalian organism can renally excrete the resulting degradation products and what consequences, if any, occur renally or in other organs. Sikora-Jasinska et al. reported a qualitatively pathological remodeling of kidney Bowman’s capsule and glomeruli, which could be related to Mo toxicity [5]. However, renal function in those animals is also known to degrade naturally with age. Biological consequences such as these need to be examined, especially for implants of larger dimensions, such as those required in craniomaxillofacial surgery.

## 4. Materials and Methods

### 4.1. Materials

If not otherwise indicated, all chemicals were purchased from Sigma-Aldrich (Deisenhofen, Germany). Cell culture materials were obtained from Greiner (Erlangen, Germany).

#### 4.1.1. Molybdenum Samples

Mo-PDMPB 1350 powder; obtained from H.C. Starck (Goslar, Germany)Mo-RSTPO80718 powder; obtained from H.C. Starck (Goslar, Germany)Mo-48Re powder; obtained from H.C. Starck (Goslar, Germany)Mo-41Re powder; obtained from H.C. Starck (Goslar, Germany)Cast molybdenum crucible obtained from KLS Martin (Tuttlingen, Germany)

#### 4.1.2. Composition and Particle Size of Molybdenum Powder

Mo-PDMPB 1350:99.5% purity;99% of particles smaller than 44 µm.Mo-RSTPO80718:99.95% purity except C and O; C and O each max. 1.1%.Particle size: max. 10% smaller than 44 µm, min. 87.5% 44–150 µm, max. 2.5% larger than 150 µm.Mo-48ReHigh-purity molybdenum powder was mixed with a 48% proportion of rhenium.Mo-41ReHigh-purity molybdenum powder was mixed with a 41% proportion of rhenium.Cast molybdenum crucibles. The molybdenum crucibles were made from high purity molybdenum powder (Mo-RSTPO80718) in a selective laser melting process and had an inner diameter of 44 mm (Figure 1).

#### 4.1.3. Cells, Fibroblast and Osteoblast Isolation and Cell Culture

For our experiments, we used cells from the NCTC clone 929 (L-929) that we purchased from LGC Standards GmbH (Wesel, Germany) and primary human skin fibroblasts (hsFB), as well as osteoblasts (hOB).

Human fibroblast cell culture: Human skin fibroblasts were isolated from human skin specimens obtained with patients’ consent from twelve female and male patients aged between 20 and 65 years who underwent plastic breast or abdominal surgery, respectively. The experimental protocol and the use of human material have been approved by the local Ethics committee of the Medical Faculty of the Heinrich-Heine-University in Düsseldorf (study number: 3634) and are in accordance with the Declaration of Helsinki.

Fibroblast cultures were prepared, cultivated and cryoconserved exactly as described elsewhere [34]. For experiments cryoconserved stocks of L929 cell or fibroblasts were thawed and further cultured (5% CO_2_, 37 °C) with cell culture media (DMEM/10% foetal bovine serum—FBS/PEN/STREP) in T 75 flasks (Cellstar, Greiner Bio-One, Frickenhausen, Germany).

For seeding, the cells were detached by two rinses with balanced phosphate buffer (PBS, pH 7.4) and incubated with 0.05% trypsin/0.02% EDTA/0.9% NaCl solution for 3 to 5 min. After cells had become detached, the trypsin activity was inactivated by addition of 10 mL culture media supplement wit 10% FBS and samples were subsequently centrifuged for 5 min at 300× *g*. Then, cells were resuspended and counted by using a Neubauer counting chamber and seeded with the cell density indicated in the respective multi-well cell culture plates one day prior starting the experimental procedure.

Human osteoblast cell culture: To establish and culture primary human osteoblast cultures, we used a method as previously described [35,36]. Briefly, cancellous bone chips were taken from the iliac crest of patients who underwent osteosynthesis with bone grafting. The three preparations used in our experiment came from two female and one male patient aged 58 ± 6 years. Exclusion criteria were known osteoporosis, corticosteroid therapy, diabetes mellitus, immunosuppression, rheumatoid arthritis, previous heparin therapy and underlying bone diseases. The minced bone fragments were cleaned after collection and transferred to a Petri dish and cultured α-MEM medium containing 10% FCS, 60 μg/mL penicillin, streptomycin, amphotericin B and 60 μg/mL ascorbic acid and cultivated in a humidified atmosphere of 95% air and 5% CO_2_ at 37 °C. The first and second passages of confluent osteoblasts were used for testing. The experiments were performed in duplicate five times.

To test a direct effect of molybdenum on the vitality behavior of primary human skin fibroblasts (hsFB) or osteoblasts (hOB), we have cultivated these cells on appropriate molybdenum crucibles. For this purpose, we used molybdenum crucibles, which were previously manufactured from high-purity molybdenum powder (SRTPO80718) in a selective laser melting process and had an inside diameter of 44 mm (Figure 1). HsFB or hOB were seeded with a cell density of 3 × 10^4^ cells/cm^2^. Experiments using hsFB were performed with cells in passage 2–4, Using hOB, experiments were performed with cells in passage 1–2.

#### 4.1.4. Preparation of Molybdenum Extracts

To produce molybdenum extracts, we mixed 20 g of the respective molybdenum powder preparations mentioned in Section 4.1.1 with 20 mL of the growth medium in a 50 mL Falcon tube and incubated the samples for 24 h at 37 °C on a roller mixer. After centrifugation of the samples at 150× *g* for 10 min the extracts (supernatants) were used in the subsequent in vitro toxicity assays.

### 4.2. Evaluation of In Vitro Cytotoxicity

The cytotoxic effects of the extracts were tested using three recognized tests: Cell-Titer-Blue (CTB) test, Neutral Red [37] test, and methyl thiazolyl tetrazolium (MTT) test. The last two tests mentioned were carried out in accordance with EN-ISO 10993-5: 2009 (Biological evaluation of medical devices—Part 5: Tests for in vitro cytotoxicity).

Briefly, in preparation for the tests, 10,000 cells per well of a 96-well cell culture plate were plated out and overlaid with growth medium (200 μL/well). The cells were allowed to adhere overnight. The supernatants were then discarded and the cells were incubated for 24 h with the prepared molybdenum extracts (200 μL/well) or, as positive control, with medium containing 1 mM H_2_O_2_ instead of the extracts. The tests were then carried out.

### 4.3. CTB-Assay

The *CellTiterBlue* assay (CellTiterBlue, Promega, Madison, WI, USA) allows to determine the relative cell numbers indirectly by quantifying the metabolic activity (complex 1 of the respiratory chain) of the respective cell cultures in the form of the ability to metabolize resazurin to resorufin. The assay was performed in accordance with the manufacturer’s protocols at time points indicated. Briefly, L929 cell samples of the test plate were incubated for 1 h with the CellTiterBlue reagent (1:20 with medium; 200 µL) and 2 × 100 µL of supernatants were taken for direct measurement at room temperature by using a fluorescence spectrometer (VICTOR II Plate Reader, PerkinElmer, Waltham, MA, USA) at an excitation wavelength of 540 nm, and an emission wavelength of 590 nm (Figure 3).

### 4.4. MTT-Assay

The MTT assay, an index of cell viability and cell growth, is based on the ability of viable cells to reduce MTT (3-(4,5-dimethylthiazol-2-yl)-2,5-diphenyl tetrazolium bromide) from a yellow, water-soluble dye to a dark blue, insoluble formazan product. Other sample plates were incubated with 200 µL MTT solution (5 g/L in PBS) for an additional 4 h, then the media were removed and 200 μL DMSO was added to each well for the following assays. The plates were shaken for 10 min or incubated at 37 °C for 15 min to obtain a sufficient extraction of the MTT products, then optical density [7] at 490 nm was detected with a 96-well microplate reader (Figure 4).

### 4.5. Neutral Red-Assay

Alternatively, the relative number of living ECs was detected by neutral red staining. Neutral red solution in phosphate-buffered saline (Sigma, Deisenhofen, Germany) was added to each well to give a final concentration of 0.03% and incubated for 90 min at 37 °C in the dark. Medium was removed, and cells were washed two times with phosphate-buffered saline and dried. After dissolving in 100 μL isopropanol plus 1% 1 M HCl, the absorption of the probes was measured at 530 nm in an enzyme linked immunosorbent assay microplate reader (Multiskan Plus MK 2, Helsinki, Finland) (Figure 5).

### 4.6. Statistical Analysis

The statistical evaluation was carried out using the Graph Pad Prism 5 and 8 software (San Diego, CA, USA). We used the paired two-tailed Student’s *t*-test or ANOVA followed by a post hoc multiple comparison test (Tukey). With a *p*-value < 0.05, the statistical differences were classified as statistically significant.

### 4.7. Immersion and Mechanical Testing of Molybdenum Sheet Metal

Pure molybdenum sheet metal of 95 µm thickness was procured from H.C. Starck (Goslar, Germany). The sheet metal was produced by rolling, followed by stress relief heat treatment. Samples for tensile testing with a total surface area of approx. 54 cm^2^ were cut from the sheet metal by electrical discharge machining (EDM). The shape of the tensile testing samples is shown in Figure 10.

Samples with two grain orientations (Figure 11) were produced by EDM: Samples with the elongated grains from rolling parallel to the axis of loading in tensile testing (left) and elongated grains perpendicular to the axis of loading (right). The grain structure of the samples was made visible by Murakami etching after metallographic preparation by grinding and polishing with alumina and diamond suspension. The left image in Figure 11 shows the microstructure parallel to the direction of rolling during manufacturing of the molybdenum sheet metal. Hot rolling, i.e., strong plastic deformation of the material at elevated temperature, induces recrystallization and leads to the formation of an anisotropic grain structure. The left image shows that hot rolling results in grains with a width of several micrometers (perpendicular to the surface) and a length of several tens of micrometers in the direction of rolling. The right image in Figure 11 shows the microstructure perpendicular to the direction of rolling. In this plane, grain sizes are small due to the preferred direction of grain growth along the direction of rolling.

Additionally, smaller samples made of the same sheet metal with a surface area of 4 cm^2^ and the same grain orientations were produced by EDM.

For immersion testing, three tensile testing samples and one smaller sample of the same grain orientation each were stored in 160 × 50 × 60 mm^3^ containers made of polypropylene. For each grain orientation, three sets of tensile testing and smaller samples were used, for periods of 2, 4 and 6 months of immersion. Sample holders were used to make sure that all surfaces of the samples were accessible to the immersion fluid. Before immersion, the samples were cleaned with 30% H_2_O_2_ to remove surface contamination and oxides formed during EDM. After cleaning, the samples were weighed on a precision scale.

The Immersion medium was Kokubo’s SBF with a pH of 7.4 and a TRIS/HCl buffer system. The ion concentrations in Kokubo’s SBF are modelled after human blood serum, except for an increased concentration of Ca^2+^ ions. For the immersion experiments in this study, the Ca^2+^ ion concentration was adjusted to the level present in human blood serum. A ratio of 2.4 mL of medium per cm^2^ of sample surface was chosen. During immersion, the sample containers were stored at 37 °C. The medium was aerated during immersion testing. The medium was changed weekly over a period of up to 6 months. Medium samples were taken during medium change. The concentration of dissolved Mo ions in used medium was measured by ICP-OES. The fluid loss due to evaporation was calculated each week by weighing the medium remaining after 7 days. This loss is taken into account when the dissolution rate of molybdenum is calculated from the ICP-OES data.

After 2, 4 and 6 months, one set each of the samples with parallel or perpendicular grain orientation was removed from the immersion fluid, dried in air and weighed on a precision scale. Cross-sections of the smaller samples were prepared for metallographic analysis by SEM and EDS. The tensile strength *R_m_* and the yield strength at 0.2% of plastic deformation *R_p_*_0.2_ of the tensile testing samples were measured on a Zwick-Roell tensile testing machine. The mechanical properties of the original molybdenum sheet metal were measured by the same method to serve as a reference.

## 5. Conclusions

In our study, we showed that pure molybdenum and molybdenum rhenium alloys do not show significant cytotoxicity in tests according to ISO standard 10993-5. The 6-month tests of the degradation of molybdenum sheets showed uniform degradation. After 6 months, there was a loss of mechanical strength of about 50%. A complete loss of mechanical strength is expected after 7.5–9 months based on these data. This would represent an optimal period for the use of an osteosynthesis material, which in most cases is removed after 6–8 months.

## Figures and Tables

**Figure 1 ijms-23-15710-f001:**
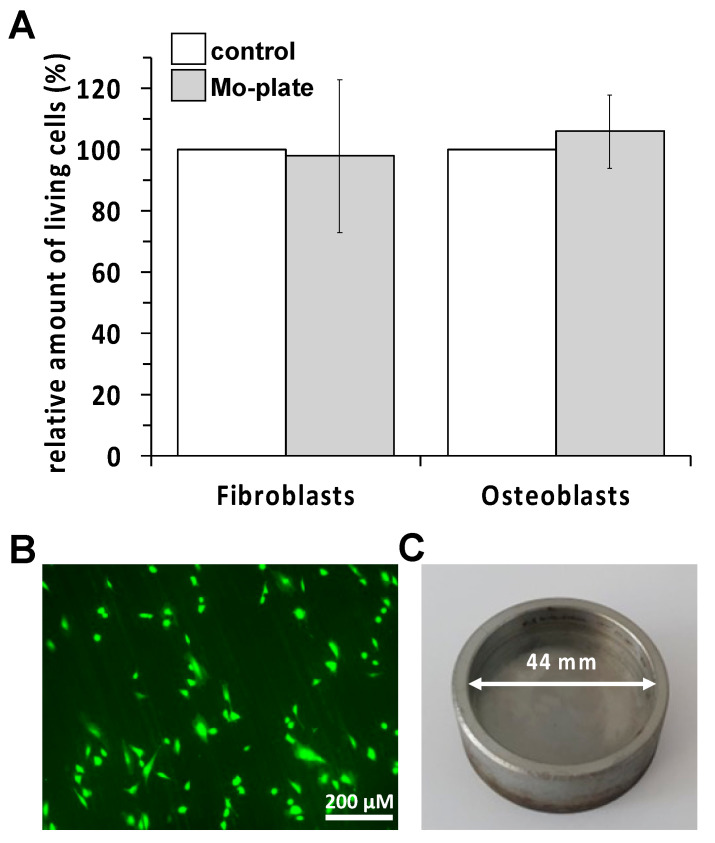
(**A**) Vitality of human skin fibroblasts and osteoblasts seeded on a molybdenum crucible in comparison to a Petri dish (control). (**B**) Positive fluorescein diacetate (FDA) staining of living adherent osteoblasts on the molybdenum crucible. (**C**) Representation of the size of the molybdenum crucible used.

**Figure 2 ijms-23-15710-f002:**
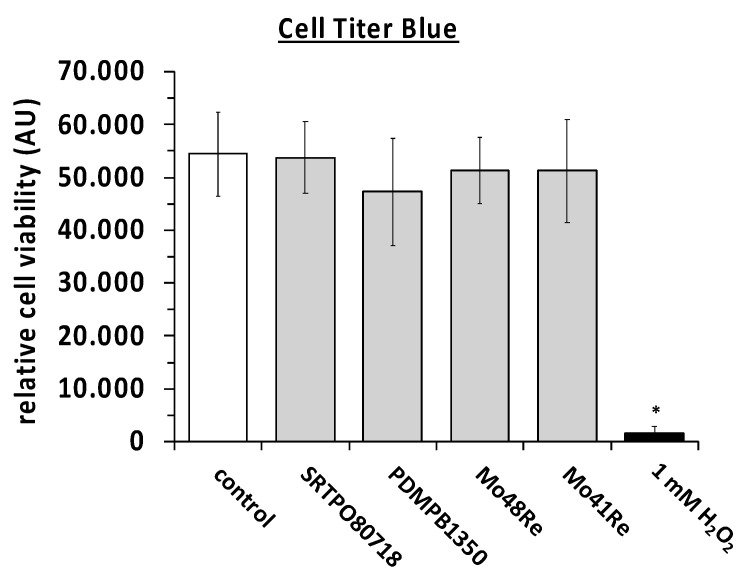
Cell titer blue assay with L929 mouse fibroblast cells *, *p* < 0.05 as compared to the control (paired two-tailed Student’s *t*-test).

**Figure 3 ijms-23-15710-f003:**
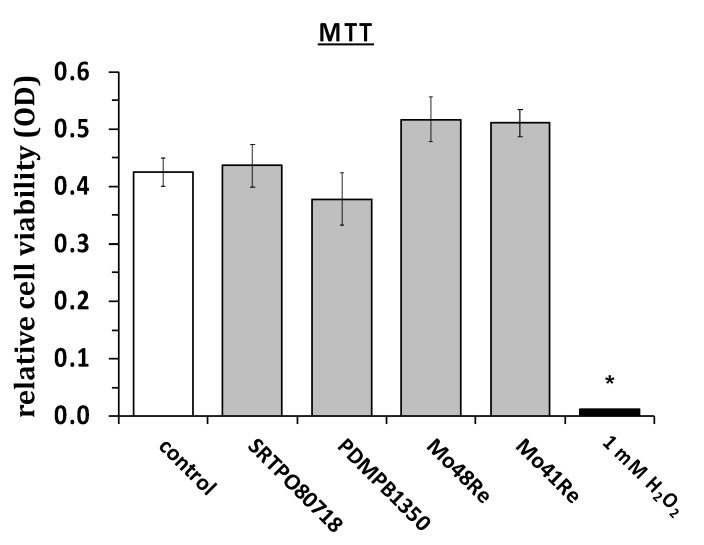
MTT assay with L929 mouse fibroblasts. *, *p* < 0.05 as compared to the control (paired two-tailed Student’s *t*-test).

**Figure 4 ijms-23-15710-f004:**
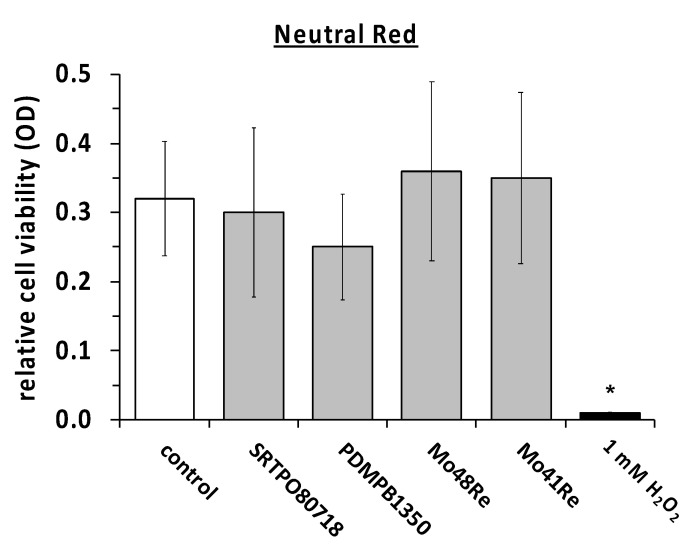
Neutral red assay with L929 mouse fibroblasts. *, *p* < 0.05 as compared to the control (paired two-tailed Student’s *t*-test).

**Figure 5 ijms-23-15710-f005:**
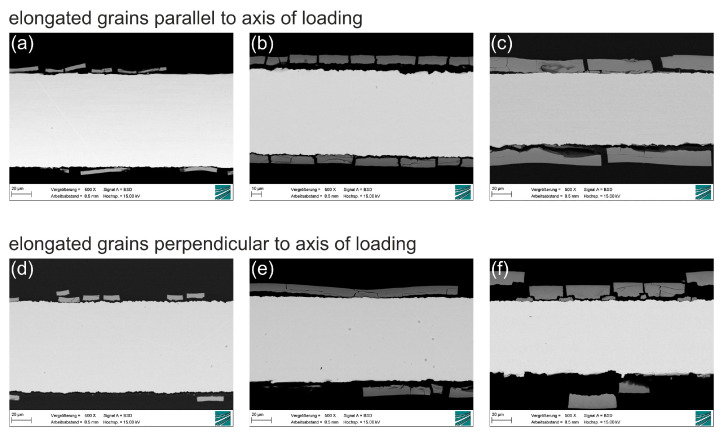
SEM images of molybdenum sheet metal after immersion for 2–6 months in Kokubo’s SBF. Images (**a**–**c**) show the sheet metal samples with elongated grains parallel to the axis of loading in tensile testing. Images (**d**–**f**) show the sheet metal samples with elongated grains perpendicular to the axis of loading (see Figure 11 in Section 4).

**Figure 6 ijms-23-15710-f006:**
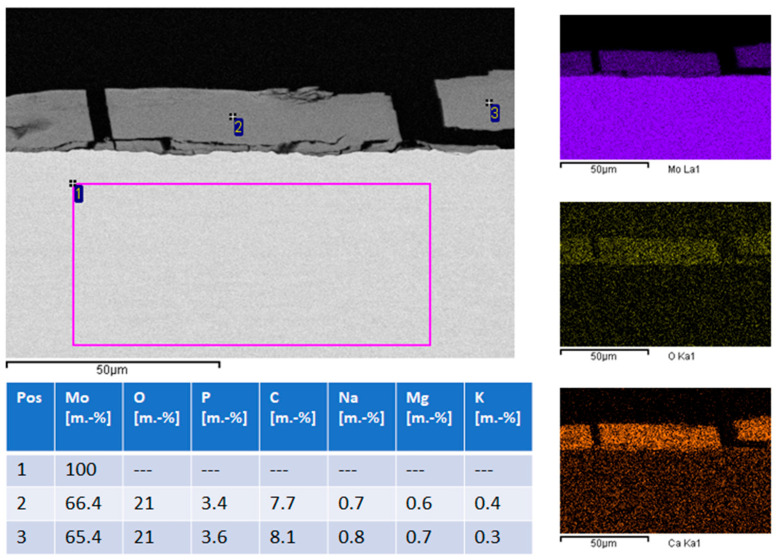
EDS (Energy dispersive X-ray spectroscopy) point analysis and mapping. The numbers 1, 2 and 3 indicate the position of the measurements. Results are given in the table. The right column of images shows the results of the EDX mapping for the elements Mo, O and Ca.

**Figure 7 ijms-23-15710-f007:**
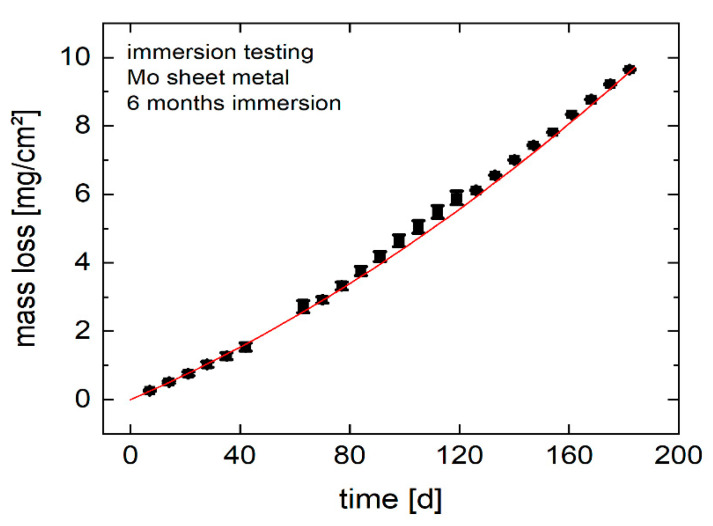
Mass loss of molybdenum sheet metal over 6 months of immersion, derived from ICP-OES measurements.

**Figure 8 ijms-23-15710-f008:**
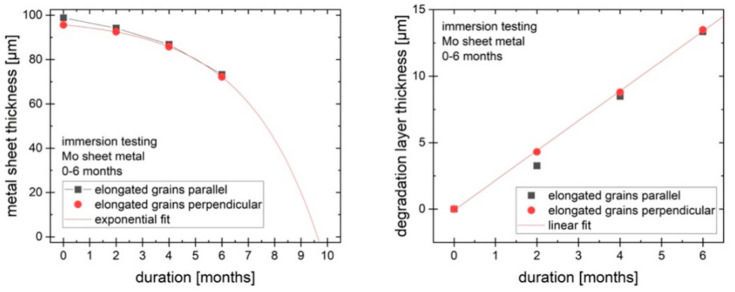
Metallic and degradation product layer thickness.

**Figure 9 ijms-23-15710-f009:**
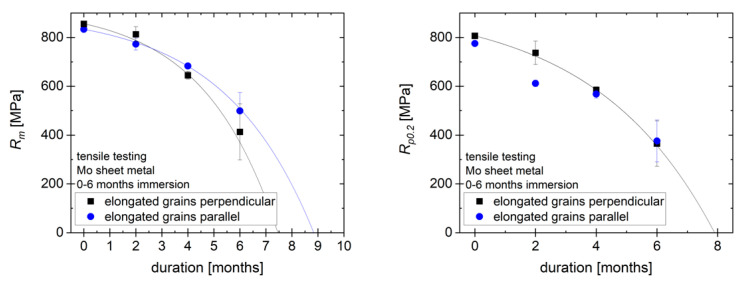
Tensile strength and yield strength of molybdenum sheet metal.

**Figure 10 ijms-23-15710-f010:**
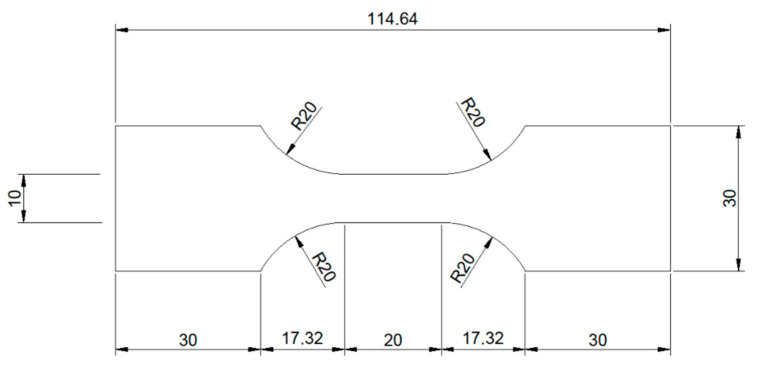
Shape and dimensions of the tensile testing samples.

**Figure 11 ijms-23-15710-f011:**
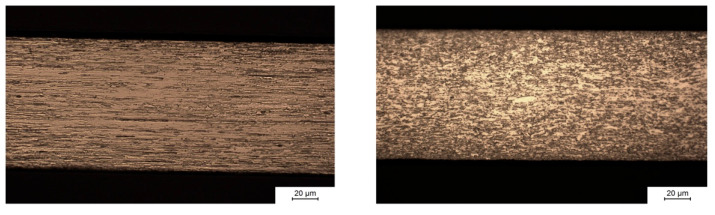
Mo sheet microstructure. Cross sections in parallel to the direction of hot rolling (**left** image) and perpendicular to the direction of hot rolling (**right** image).

**Table 1 ijms-23-15710-t001:** Total mass loss and area-specific mass loss of molybdenum sheet metal samples in 2–6 months of immersion in Kokubo’s SBF.

Grain Orientation	Duration of Immersion (Months)	Total Mass Loss (%)	Area Specific Mass Loss (mg/cm^2^)
perpendicular	2	7.6	3.8
perpendicular	4	13.0	6.4
perpendicular	6	19.8	9.6
parallel	2	6.0	3.0
parallel	4	12.2	6.0
parallel	6	19.3	9.5

## Data Availability

For further information, please contact the corresponding author.

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
