# Peer review of "Molybdenum as a Potential Biocompatible and Resorbable Material for Osteosynthesis in Craniomaxillofacial Surgery—An In Vitro Study"

_ijms, 2022, doi:10.3390/ijms232415710_

Round 1
Reviewer 1 Report
The Mo degradation and biocompatibility are attracting more attention, the manuscript showed some new results on them, which is interesting. However the study is a little simple and the manuscript writing should be improved. Some questions were shown in the following for your discussion.
1. Can the Mo-Re alloy be absorbed in vivo as the Re content is very high?
2. Why was the Mo-Re alloy used in some experiments of this study?
3. How are the degradation products absorbed or degraded? Can they be absorbed completely?
4. The part of materials and methods should be descript with more details. Such as in 4.1.4, what were the molybdenum preparations used in the study? Plates or powder? What are the samples in 4.1.1?
Reviewer 2 Report
This study investigated the degradability and biocompatibility of Mo or its alloys as new biodegradable materials for bone use. It showed novelty while many issues should be clarified.
Materials & Methods
1. 5 kinds of Mo alloys were used, while their form or state is not clear. Are they all powders? What is the difference between them. There was only 1 table presenting the chemical composittions and physical properties.
2. Cast Mo plate was used. The preparation or casting process should be provided in detail.
3. The microstruture of as-rolled or heat treated Mo alloys were presented. Some explanation should be provided to tell the readers about information of each image.
Results and Discussion
1. Figure 1, what is the metallic plate in figure 1. What is the state of this plate?
2. statistical analysis should be provided in Fig. 2-4.
3. I can not differentiate the grains or elongated grains in figure 5. Was it etched?
4. In fig. 8, is it valible to simulate the drop of plate thickness with only 4 data points. As Mo degradation was rather slow from the current results, even slower than Zinc alloys. Is it possible to degrade completely or loss the total plate thickness after 10 months immersion?
5. The authors described the osteosynthesis properties of Mo alloys. But no osteoblasts or related cell lines were used in this study.
Round 2
Reviewer 1 Report
It can be accepted in the present form.
Reviewer 2 Report
The manuscript has been sufficiently improved.